# The Impact of COVID-19 Pandemic on Female Sexual Function

**DOI:** 10.3390/ijerph19063349

**Published:** 2022-03-12

**Authors:** Anna Fuchs, Mateusz Szul, Agnieszka Dulska, Jakub Bodziony, Aleksandra Matonóg, Joanna Pilarska, Paulina Sieradzka, Rafał Stojko, Agnieszka Drosdzol-Cop

**Affiliations:** Chair and Department of Gynecology, Obstetrics and Oncological Gynecology, Medical University of Silesia in Katowice, 40-211 Katowice, Poland; mszul100@gmail.com (M.S.); dulska.agnieszkaz@gmail.com (A.D.); jt.bodziony@gmail.com (J.B.); aleksandra.matonog@gmail.com (A.M.); yoasiapil26@gmail.com (J.P.); pausieradzka@gmail.com (P.S.); rstojko@sum.edu.pl (R.S.); cor111@poczta.onet.pl (A.D.-C.)

**Keywords:** COVID-19, COVID-19 pandemic, sexuality, sexual dysfunction, female sexual function

## Abstract

Introduction: The outbreak of the COVID-19 pandemic affected many aspects of life involving sexual functioning. Methods: This prospective, noninterventional, observational research was performed from July 2020 to August 2020, involving a total of 644 patients. Female sexual function index (FSFI) scores of those women were compared in relation to results of our research from April 2020. Questionnaires were collected concerning socio-demographic characteristics of patients as well as the impact of the COVID-19 pandemic on their lives. Results: Every domain significantly decreased in the first month of the lockdown. Before the pandemic, the total FSFI was reported at 30.1 ± 4.4, while in the first month of the pandemic it was at 25.8 ± 9.7 (*p* < 0.001). The lowest FSFI score was reported by women who lived in average conditions. There was a statistically significant increase in the overall FSFI score in the fifth month of the pandemic (27.5 ± 6.8) when compared to the FSFI score in the first month of the pandemic (25.8 ± 9.7). The scores in arousal, lubrication, orgasm and pain were reported as higher (respectively *p* < 0.001, *p* < 0.001, *p* < 0.001, *p* < 0.05), while desire was reported lower (*p* < 0.001). The satisfaction score remained almost the same (*p* < 0.05). Conclusions: Our study indicates a significant decrease in FSFI scores across all domains. There are still many questions whether the statue from the vaccinated person or convalescents affects the sense of security and, thus, increases sexual satisfaction.

## 1. Introduction

The year of 2020 reworked the definition of personal space and intimacy. Working online, social distancing, masks and gloves became the new reality. It was meant to ensure our health and wellbeing, but no one predicted its influence on a sense of safety and security. The rate of depression in the world population in 2017 was 3.4%, but the latest data claim that the prevalence of this mental illness during the lockdown could be even seven times higher. The first weeks and months of the pandemic had an especially exceptional impact on the human psyche, because at the beginning of the lockdown, people experienced unexpected changes in their lives. Additionally, severe acute respiratory syndrome coronavirus 2 (SARS-CoV-2) was still unfathomable and the unknown always causes the greatest anxiety. As the coronavirus disease 2019 (COVID-19) pandemic continued, the government resigned from further restrictions, public places were opened, and many people returned to stationary work and normal habits. Life slowly began to resemble that of before the pandemic and people began to become used to the new reality [1].

Severe acute respiratory syndrome coronavirus 2 is a pathogen that causes the COVID-19 disease, which was first identified in Wuhan, Hubei, China, in December 2019. COVID-19 affects different people in different ways. Symptoms might be mild, moderate and severe, among which dry cough, fever and tiredness are considered the most common. Additionally, patients can present less common symptoms such as aches and pains, a sore throat, diarrhoea, conjunctivitis, headache, the loss of taste or smell, a rash on the skin or the discolouration of fingers or toes. The outbreak of the COVID-19 pandemic surely affected many aspects of life worldwide. During the first months, places such as restaurants, bars and clubs were closed and people were especially deprived of opportunities to meet someone new, develop friendships and become partners. The rule of keeping distance did not help either [2,3,4].

The first case of laboratory confirmed SARS-CoV-2 in Poland was announced on 4th March 2020. On 10–12 March, in response to the growing number of new infections in Poland, new rules were introduced, including the closure of many public spaces, limiting interpersonal contacts and announcing a mandatory 14-day quarantine for people returning from abroad and after contact with the infected [5].

As the COVID-19 pandemic continued, the government resigned from further restrictions, public places were reopened and many people returned to stationary work and normal habits. While people were slowly becoming used to social distancing, life slowly began to resemble the one from before the pandemic [6].

However, the impact of isolation on mental health is definitely long-term and also involves sexual functioning. Sexuality is an extremely important aspect of every woman’s life. Unfortunately, it is still often neglected these days. Sexuality is a complex process, coordinated by the nervous, vascular and endocrine systems. Individually, sexuality incorporates familial, societal and religious beliefs, and is altered with aging, the health status and personal experience. To objectively measure sexual satisfaction, scales such as the female sexual function index (FSFI) were created [7,8].

Our study aims to investigate the impact of COVID-19 pandemic restrictions on female sexual functioning in Poland just after the initial lockdown and five months later, comparing both of them.

## 2. Materials and Methods

This prospective, noninterventional, observational research was conducted from 25 July 2020 to 10 August 2020, involving a total of 644 patients. Female sexual function index (FSFI) scores of those women were compared in relation to results of our research from April 2020. Both were carried out at the Department of Pregnancy Pathology, Department of Woman’s Health, School of Health Sciences in Katowice, Medical University of Silesia, Poland.

The study was conducted with sexually active participants of reproductive age. The main exclusion criteria was the presence of sexual dysfunctions, lack of sexual activity, age under eighteen, using medicine reducing libido in the last three months, personality disorders or other mental illnesses, including depression, marital conflicts and positive test for COVID-19.

To evaluate sexual function, FSFI questionnaire was used, which is a self-report survey of 19 items in 6 subscales: desire, arousal, lubrication, orgasm, satisfaction and lack of pain. The total scale was achieved by adding the scores obtained in the 6 domains. The maximum score was 36 and minimum was 2. Dysfunction was recognized at value of 26 and below. The FSFI assesses sexual functioning over the past 4 weeks. First surveys were conducted at the beginning of March, before the first case of COVID-19 in Poland when there were no government restrictions. The second questionnaire was gathered in the second half of April, while the Polish government had forced the population to be quarantined at home and restrictions were the strongest thus far. Other data were collected at the turn of July and August, in the fifth month of the pandemic, while in Poland the lockdown had been lifted. In previous research, the questionnaires were filled in by 764 women; unfortunately, 120 of them refused re-participation in the study [9,10].

The university Ethics Committee waived the requirement for informed consent due to the noninterventional nature of the study (PCN/0022/KB1/108/20). Patients gave informed consent to the study.

StatSoftStatistica version 13.0 PL software (Dell, TX, USA) was utilized to conduct all data analyses. *p*-value < 0.05 was considered as significant. Analyses of dependent variables were evaluated by ANOVA Fridman and the Wilcoxon’s rank test. Regarding quantitative variables, comparison Chiˆ2 was used. Kruskal–Wallis test was implemented for independent variables, and an additional inquiry was executed by the Mann–Whitney U test. Mean ± SE (standard error) presents all of the data.

## 3. Results

In total, 644 patients participated in the questionnaire. The age of the study participants ranged between 19 and 40 (26.5 ± 4.5). Approximately a third (*n* = 220) of the participants questioned lived in cities with over 500,000 residents, a third (*n* = 237) in cities with over 50,000 residents and a third in cities, towns and villages with less than 50,000 residents (*n* = 187). Most of the women were in informal relationships (71.1%), 37.9% were married, while 7.3% of them did not have a partner. Almost half of the respondents (48.1%) determined their living conditions as good, 36.7 % as very good and 15.2% as average. Over two-thirds of respondents had never been pregnant before (73%). Approximately a quarter of the questioned participants (*n* = 161) already had one child and 13 of them were actually pregnant. Sociodemographic characteristics of participants are presented in Table 1.

As presented in Table 2, every domain significantly decreased in the first month of the lockdown in comparison to the time from before the pandemic. Before the outbreak, the total FSFI was reported at 30.1 ± 4.4, while in the first month of the pandemic, it was at 25.8 ± 9.7 (*p* < 0.001) (Table 2).

In the fifth month of the lockdown, in comparison to the time before the pandemic, the same tendency was observed. Before the outbreak, the total FSFI was reported at 30.1 ± 4.4, while in the fifth month of the pandemic, it was at 27.5 ± 6.8 (*p* < 0.001) (Table 2).

There was a statistically significant increase in the overall FSFI score in the fifth month of the pandemic (27.5 ± 6.8) when compared to the FSFI score in the first month of the pandemic (25.8 ± 9.7). The scores in arousal, lubrication, orgasm and pain were reported as higher (respectively, *p* < 0.001, *p* < 0.001, *p* < 0.001 and *p* < 0.05), while desire was reported lower (*p* < 0.001). The satisfaction score remained almost the same (*p* < 0.05) (Table 2).

The amount of sexual intercourse also varied over time and was the highest before the lockdown, when every woman was sexually active. At the beginning of the COVID-19 pandemic, as much as 15.2% of women reported not having intercourse, while in the fifth month, it was only 3.9%. Women’s sexual activity depended on the stage of the pandemic and is revealed in Figure 1.

The results of our study demonstrated that there was no significant association (*p* > 0.05) between the age, number of pregnancies, marital status, education and place of residence, and the overall FSFI score in the fifth month of the pandemic.

The lowest FSFI score was reported by women who lived in average conditions (25.4 +/− 8.2) in comparison to both good (28.1 +/− 5.8; *p* < 0.001) and very good (28.0 +/− 7.1; *p* < 0.001).

## 4. Discussion

The main aim of our study was to compare the FSFI score before and during the first and the fifth months of the COVID-19 pandemic to establish if there were differences depending on the time of becoming used to this extraordinary situation. To our knowledge, this was the first study which compared these three periods of time [11].

The outbreak of the COVID-19 pandemic was unexpected and one had very little time to adjust to a radically new reality. It would seem that a greater amount of time spent with a partner and a slower pace of life, both caused by the forced stay-at-home, should have led to the tightening of the bond between partners and to an increased intimacy in the relationship. In fact, a study by Cito et al. (2021) showed a significant increase in the amount of sexual intercourse during the first weeks of lockdown [12]. The values, however, began to drop after two weeks of quarantine [12,13].

This behaviour may be related to concerns about being infected with the virus. The ubiquitous information about the possibility of contracting an infection from an asymptomatic patient, inaccurate information about the routes of transmission of the virus at the beginning of restrictions, general misinformation and the lack of contact with their doctors could lead to fear of sexual intercourse [14,15]. Moreover, messages from high-authority centres did not reassure patients. For example, at the beginning of the restrictions on the Mayo Clinic website, there was information about safe sexual intercourse. It was recommended to minimize the number of sexual partners to as few as possible, to avoid partners with symptoms of an upper respiratory tract infection and to avoid kissing. Moreover, recommendations regarding the use of masks during sexual intercourse certainly did not increase the desire for sex [16].

Our study showed a significant decrease in FSFI scores across all domains. Similar results were obtained in Italy in a study by Schiavi et al. (2020) Researchers associated a decline in FSFI scores with patients’ overall fear of infection [17]. The first weeks of the lockdown significantly influenced the appearance of symptoms, such as a depressed mood, anxiety and depression. People who were used to living in a constant rush and to leading an active lifestyle were deprived of multiple stimuli almost overnight and began to show distress. The beginning of the pandemic meant an increase in the number of employees working remotely. It became clear that prolonged isolation, working at home, constant negative news from the media and the fear of what the next weeks would bring had a negative impact on the mental health of people all over the world. Moreover, the lack of knowledge of the expected end of the pandemic and the possibility of the virus remaining forever led to restlessness. The patients who were already suffering from depression and anxiety began to report the exacerbation of their previously stabilized disease. At the same time, symptoms of depression appeared in many previously healthy patients [18,19].

Constantini et al. performed research in Italy, where an improvement in sex life was observed in participants who lived with their partners. In people who did not live in the same household, the opposite trend was observed, although with no statistical significance. However, Italians are known for their sense of community and, thus, deprived of social life, they focused on home life. The results were very interesting, especially since the situation of the COVID-19 outbreak was a major issue. The study included both women and men [20].

We did not analyse psychological factors, and the exclusion criteria of our study were mental illnesses, including depression. However, an interesting study by Mollaioli et al. highlights that people who remained sexually active during the lockdowns had significantly lower scores of anxiety and depression [21]. It seems that sexual activity plays an important role as a protective factor in both genders on mental illnesses. On the other hand, Ilgen et al. also observed an increase in anxiety and depression scores, but with no impact on female sexual behaviour [22]. The mentioned field, however, needs further research.

All these factors can negatively affect sexual function. In addition, a prolonged stay at home without going outside in many cases also led to a reduction in care for one’s appearance, which may also be a factor contributing to the reduction in the feeling of attractiveness and sexual activity [23,24]. The study by Luis et al. (2021) proved that the higher the perceived stress of the situation during the COVID-19 pandemic, the more difficult it was to take part in self-care activities [24]. Hence, it had implications for personal growth and psychological well-being as well. Those may be connected with one’s lower self-esteem and could, eventually, also affect sexual functioning and satisfaction of sex life [24]. Additionally, research conducted by Karagöz et al. (2020) in Turkey also showed a significant drop in all of the domains of the FSFI score [25]. The participants showed significantly higher levels of anxiety, a high level of perceived stress and symptoms of depression, particularly noticeable among women [25].

Staying at home had a positive impact on the closeness of the relationship. Yet, we have to remember that the household is often not a place only for the ‘two of us’. Many people gave up sexual intercourse due to the lack of intimacy. This applies not only to the families with children, but also to the young people living under the same roof with their parents or other residents. In addition, restrictions such as prohibition to leave the place of residence could significantly hinder contact with a partner [26]. On the other hand, the lack of ability to meet the partner might sometimes have a good effect on relationships, especially those where boredom in terms of sex life has occurred. Lopes et al. (2020) investigated those who struggle with living apart from steady partners, and suggested reinventing sexual life with adding alternatives such as sexting, playing sexual games and sharing photographs and videos [27]. All of them may awaken one’s desire with psychological, visual and auditory stimulation while not requiring physical contact. It can also provide a new experience and help discover each other’s fantasies [27].

An important factor influencing sexual satisfaction is certainly the sense of security. Of course, the onset of the pandemic created a sense of an ubiquitous threat of infection. At the same time, all medical forces were directed to fight coronavirus, which made it much more difficult for patients to access basic medical care and specialists [28]. In addition, shipping of many drugs was blocked from countries such as China and India. We know from clinical practice that many women in Poland lived in fear of becoming pregnant due to the shortage of contraceptive drugs used. We did not ask about contraception in our study, but we cannot exclude that this issue may have influenced the obtained results.

Although similar conclusions were drawn in the United States, it should be remembered that many countries did not struggle with this problem. For example, in France, there was an increase in the use of contraceptive pills when compared to the pre-pandemic period. However, the use of contraceptive methods requiring direct contact with health care professionals (LNG-IUDs, ovulation inductors) decreased significantly [29,30].

The significant association was found between living conditions and the total FSFI score. The lowest FSFI score was observed among women who lived in average conditions, when comparing the group with good and very good conditions. We all remember the sudden increase in the price of personal protective equipment at the start of the pandemic. Gloves and masks became a luxury product then. Especially during the first months, it could have had a significant impact on the sense of security of people in a worse financial situation. In addition, people in worse financial situations had to use public transport more often, exposing themselves to contact with potentially infected people [30,31,32]. It seems that having less money and living in worse conditions means more stress and anxiety, which impacts sexual functioning.

After a significant drop in the FSFI score during the lockdown, we observed a re-increase after a few months. This also applied to the frequency of intercourse. This is extremely important because among both women and men sexual intercourse is positively linked to higher levels of relationship satisfaction, as well as physical and emotional wellbeing [33]. After the first few months of this stressful situation, people began to adapt to a previously unknown way of life. In addition, further assurances about the soon-to-emerge vaccinations were optimistic. Becoming used to this strange situation certainly had a positive effect on the increase in the frequency of intercourse and the slow pursuit of regaining sexual satisfaction from before the pandemic.

## 5. Conclusions

The main finding of our study was the important influence of the COVID-19 pandemic on sexuality among Polish women. Our previous study showed a significant decrease in FSFI scores across all domains in the first month of the pandemic compared to the time before the pandemic. After five months, they all seemed to re-increase compared to the results received at the beginning of the pandemic, which was related to the adaptation to the new reality. An interesting issue seemed to be the state of sexual satisfaction in a situation, in which we have available weapons in the form of vaccines, which were introduced in the beginning of 2021 in Poland. There are still many questions such as whether the statue from vaccinated persons or convalescents affects the sense of security and, thus, increases sexual satisfaction, so further research is required [34,35].

## Figures and Tables

**Figure 1 ijerph-19-03349-f001:**
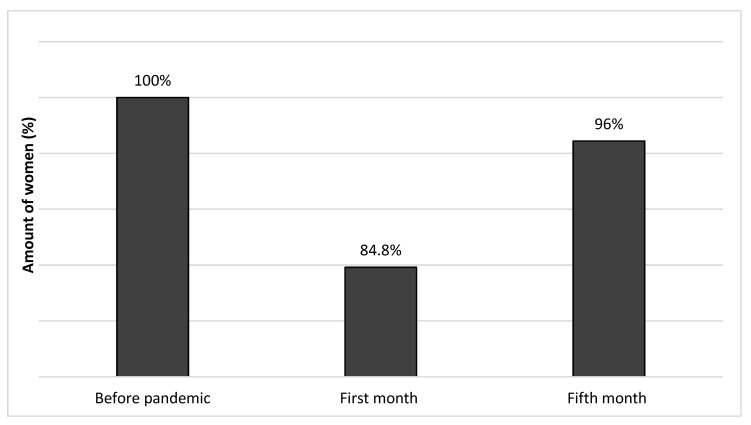
Percentage of sexual intercourse before, in the first month and in the fifth month of the pandemic.

**Table 1 ijerph-19-03349-t001:** Demographic characteristics of the study participants.

Characteristics	
**Total respondents:**	**644**
Age (years)	19–40

Average	26.5 +/− 4.5
**Marital Status**	

Informal relationship	458

Married	139

Single	47
**Education**	

Primary	4

Secondary	157

Higher	483
**Place of residence**	

City with over 500,000 residents	220

City with 250,000–500,000 residents	119

City with 50,000–250,000 residents	118

Town with less than 50,000 residents	82

Village	105
**Living conditions**	

Very good	236

Good	310

Average	98
**Pregnancy**	

Never	470

Once	79

More than once	82

Actually	13

**Table 2 ijerph-19-03349-t002:** Comparison of FSFI scores before, in the first and in the fifth month of the COVID-19 pandemic. Statistical significance was set at *p* < 0.05.

Variables	1 Before the Pandemic	2 First Month of the Pandemic	3 Fifth Month of the Pandemic	1 vs. 2	1 vs. 3	2 vs. 3
Desire	4.5 ± 1.0	4.2 ± 1.3	3.9 ± 1.2	<0.001	<0.001	<0.001
Arousal	5.1 ± 0.9	4.1 ± 2.0	4.5 ± 1.4	<0.001	<0.001	<0.001
Lubrication	5.4 ± 0.8	4.5 ± 2.1	5.1 ± 1.4	<0.001	<0.001	<0.001
Orgasm	4.8 ± 1.3	3.9 ± 2.1	4.4 ± 1.6	<0.001	<0.001	<0.001
Satisfaction	5.2 ± 1.0	4.7 ± 1.4	4.7 ± 1.3	<0.001	<0.001	>0.05
Pain	5.1 ± 1.1	4.3 ± 2.1	4.9 ± 1.5	<0.001	<0.05	<0.001
Total FSFI	30.1 ± 4.4	25.8 ± 9.7	27.5 ± 6.8	<0.001	<0.001	<0.05

## Data Availability

Not applicable.

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
