# Peer review of "The Impact of COVID-19 Pandemic on Female Sexual Function"

_ijerph, 2022, doi:10.3390/ijerph19063349_

Round 1

Reviewer 1 Report

The manuscript "The impact of COVID-19 pandemic on female sexual function." in an observational study that evaluates the impact of lockdowns on female sexual function as measured with an index that assesses different aspects of the female sexual function and that can be used to diagnose sexual disfunction in women. 

A major issue through the manuscript is the quality of the english language, which requires some significant revision to be made fully understandable. A minor issue was the incorrect format for citations, especially in the discussion, where numberes wer enot used and some references where cited by making reference to the author and year. For example, in lines 95-97 where successive apparently contradictory statements make unclear if the patient consent was requested and the requirement of the institutional review board.

Other than that the study seems to have been well performed, the sample size was large enough and reporting seems to be in accordance with STROBE requirements for observational epidemiological studies. The data support the conclusions and overall the data is very illustrative of .one of the many aspects by which some aspects of female population health were affected.

Author Response

Thank you for your time and all your valuable remarks. We are extremely grateful for such favorable feedback. We improved the English following your advice, as well as we performed necessary changes in citations. 

Reviewer 2 Report

Dear Authors,

Sorry for the delay in submitting this review.

Congratulations for your work. The study is interesting and (to an extent, see below) adequately designed.

There are a few issues, however, which I believe should be taken into consideration.

  1. I'm a bit surprised by the missing information on the psychological status of people included in the study, also given the fact that the relevance of psychological conditions is mentioned several times in the manuscript.
  2. I perfectly agree with the choice of using a young sample - however I would add a brief mention of the non-generalizability to the overall population.
  3. On a side note, there is some evidence suggesting that cardiovascular health (a key determinant of male sexual function) can also be associated with female sexual function. The role of COVID-19 on cardiovascular health can be relevant for this. Maybe it might be worth looking into this? Smoking habits, BMI and use of medications for CVDs could be other factors worth investigating.

Author Response

Thank you for your time and all your valuable remarks. We allowed ourselves to present the changes and responses to individual comments in points.

  1. Thank you very much for your valuable tips. Unfortunately, in this study we did not focus on the psychological status of the patients. The onset of the pandemic has not yet indicated such major changes in the mental health of the population. However, this topic is of great interest to us and we analyzed it in our further research on female sexuality and the COVID 19 pandemic. As we highlighted in material and methods section - the exclusion criteria in this paper were personality disorders or other mental illness including depression. We do not believe that the information on mental health in the first months of the pandemic has a significant impact on the results obtained.
  2. As we indicated in the materials and methods, the surveyed women were of adult, reproductive age and sexually active. Due to the nature of our clinic, patients who met these criteria were in the 18-40 age group. It is in this age group that the patients willing to participate in the study were found. Moreover, in this study, the Patient group was imposed in advance. It is a follow-up and in our previous work these were the inclusion criteria.

  3. Thank you for your good remark. Unfortunately, we are unable to verify the parameters you mentioned. However, this remark is extremely valuable, and we will certainly follow it in future research.

Reviewer 3 Report

Thank you for the opportunity to review this manuscript that aim to evaluate the impact of covid measures in female sexual life. To date there are several manuscripts reporting these data, and several have been reported in the bibliography. What make this manuscript interesting is the sub analysis reported in table 1, and in particular the marital status. However, in the discussion would have been interesting to expand the discussion also based on the type of relationship. This point should be really improved. I suggest authors to read, and if considered useful to report "The impact of lockdown on couples' sex lives. Costantini et al. J Clin Med 2021 Apr 1;10(7):1414. doi: 10.3390/jcm10071414."

More suggested papers are: 

Sexual Health Implications of COVID-19 Pandemic. Pennanen-Iire C, Prereira-Lourenço M, Padoa A, Ribeirinho A, Samico A, Gressler M, Jatoi NA, Mehrad M, Girard A. Sex Med Rev. 2021 Jan;9(1):3-14. doi: 10.1016/j.sxmr.2020.10.004. Epub 2020 Nov 5.PMID: 33309005

Benefits of Sexual Activity on Psychological, Relational, and Sexual Health During the COVID-19 Breakout. Mollaioli D, Sansone A, Ciocca G, Limoncin E, Colonnello E, Di Lorenzo G, Jannini EA.J Sex Med. 2021 Jan;18(1):35-49. doi: 10.1016/j.jsxm.2020.10.008. Epub 2020 Oct 23. 

COVID 19sexual vulnerabilities and gender perspectives in Latin America. Silveira Campos L, Brigagão de Oliveira M, Peixoto Caldas JM.Health Care Women Int. 2020 Nov-Dec;41(11-12):1207-1209. doi: 10.1080/07399332.2020.1833884. Epub 2020 Nov 10.

COVID-19 pandemic effect on female sexual function. Ilgen O, Kurt S, Aydin C, Bilen E, Kula H.Ginekol Pol. 2021;92(12):856-859. doi: 10.5603/GP.a2021.0084. Epub 2021 Apr 29.

COVID-19 and the Changes in the Sexual Behavior of Men Who Have Sex With Men: Results of an Online Survey. Shilo G, Mor Z.J Sex Med. 2020 Oct;17(10):1827-1834. doi: 10.1016/j.jsxm.2020.07.085. Epub 2020 Aug 10.

If authors are interested it would be possible (and fancy) to find some issues also in gay-relationship showing how some mechanisms in relationship per se are more similar also with males.

Last but not least authors did not mention limits of the study. All studies have limitations and one that I suggest to report and argue is the difference in cultural attitudes, thus also in sexual life, in the different populations. Obviously authors reported the picture of female sexual live in Poland during the pandemic CoVid chaos. But the comparison with other manuscripts may be affected by these cultural limitations.

Author Response

Thank you for your time and all your valuable remarks. We are extremely grateful for such favorable feedback. Thank you for your great advices. We followed them, performed extensive research and improved the discussion part. Cosidering the gay - relationship the subject is very interesting, however, we would like to focus on female sexuality. It is a great idea for another research though. Thank you for your point on the limitations. As we indicated in the materials and methods, the surveyed women were of adult, reproductive age and sexually active. Due to the nature of our clinic, patients who met these criteria were in the 18-40 age group. It is in this age group that the patients willing to participate in the study were found. Moreover, in this study, the Patient group was determinated in advance (because it is a follow-up) and in our previous work these were the inclusion criteria. We believe that the limit of the study is presented in the inclusion criteria.

Reviewer 4 Report

The authors investigate sexual functioning changes before and during the pandemic in a sample of 644 women.  The paper needs careful reviewing for English language conventions, vocabulary, and grammar.  I provided just a few examples in the abstract, but did not review the whole paper for editorial issues.  The literature review provides a good background on the emergence of COVID-19 generally and specific to Poland, however there is no background on sexuality, sexual behaviour, or sexual functioning in relation to COVID-19. There are other published papers on these topics which would provide a background for the current work and should be included.  Need much more information about recruitment.  Where were participants sought?  What were they told the study was about?  Was the study always intended to be longitudinal?  Or were participants recontacted after to determine the impact of the pandemic?  Where did they complete the questionnaires (on site or online?)?  What other measures did they complete?  Was this part of a larger study?  The results are clear and the discussion is interesting.  More citations could be added to support ideas, as there has been a wealth of research published on sexuality and relationships during the pandemic.  The authors analyzed whether marital status, age, etc. were associated with FSFI scores in the 5th month but they are not making the most of their data.  Did those factors impact the degree of decline after the pandemic?   The authors should seek advice about how to use those moderators in their analyses to determine their impact. Nonetheless, the paper is on a very important topic and these findings will be useful in the literature for understanding the impact of the pandemic on sexual functioning.  These changes can be fairly easily integrated.

Editorial comments

  • Abstract line 10 - delete "the" before sexual functioning
  • Abstract line 12 - score should be plural (scores)
  • Abstract line 17 - delete comma before who
  • Abstract line 22 - rather than "showed" say "indicated"
  • Abstract line 23 - the last sentence doesn't make sense, statue from the vaccinated person?  also convalescents?
  • page 3 line 72 - say "conducted" rather than "performed"
  • page 3 line 78 - of reproductive age (not in)
  • page 3 line 79 - write in past tense, "was the presence of sexual dysfunction"
  • page 6 line 169 - move this up so it's part of the same paragraph as the results about sexual functioning 
  • page 6 line 182 - anecdotally we have heard that people stopped caring about their appearance in the pandemic but to make this claim in a peer-reviewed journal article you need to have a reference to back this up.  It would be great if you had one, though, as this is such a common belief, it would be great to actually document it.
  • page 7 lien 221-228 - be more explicit about the potential financial difficulty and sexual functioning - having less money also means more stress and anxiety and this would impact sexual functioning 

Author Response

Thank you very much for all your comments. We are extremely grateful for such favorable feedback. Following your opinion we have extended the discussion. We included more sexuality - related background.. We improved the text following your advice about editorial and language changes. 

Answering questions regarding the recruitment: The participants were our patients from our clinics and they were sexually active woman in reproductive age without sexual disfunctions. They were told that our study was about the impact of COVID-19 pandemic on the women sexual health. Our first study was meant to investigate the impact of social quarantine on women sexual health. While collecting those surveys it wasn’t known how long COVID-19 pandemic would last, so it was hard to plan whether and when the study would be continued. After some months of the duration of the pandemic and quarantine we came up with an idea to perform the study once again. Thanks to the fact that participants were our clinic's patients it was easier to contact them via email again - all of the questionnaires were collected via email. In the previous research the questionnaires were filled in by 764 women, unfortunately 120 of them refused re-participation in the study. Except for Female Sexual Functioning Index and number of sexual intercources we also asked participitiants about exclusion criteria: presence of sexual dysfunctions, lack of sexual activity, age under eighteen years old, established diagnosis of COVID-19, using medicine reducing libido for last three months, mental illness including depression or personality disorders and marital conflicts. Our study is a part of a larger research, it is a follow-up of our previous article (https://doi.org/10.3390/ijerph17197152).

Round 2

Reviewer 3 Report

Authors improved the manuscript. Thus, I suggest the Editor to accept the manuscript.

This manuscript is a resubmission of an earlier submission. The following is a list of the peer review reports and author responses from that submission.